# Position: Machine Learning Models Have a Supply Chain Problem

**Sarah Meiklejohn** [1]  **Hayden Blauzvern** [1]  **Mihai Maruseac** [1]  **Spencer Schrock** [1]  **Laurent Simon** [2]
**Ilia Shumailov** [2]

## Abstract

Powerful machine learning (ML) models are now readily available online, which creates exciting possibilities for users who lack the deep technical expertise or substantial computing resources needed to develop them. On the other hand, this type of open ecosystem comes with many risks. In this paper, we argue that the current ecosystem for open ML models contains significant *supply-chain* risks, some of which have been exploited already in real attacks. These include an attacker replacing a model with something malicious (e.g., malware), or a model being trained using a vulnerable version of a framework or on restricted or poisoned data. We then explore how Sigstore, a solution designed to bring transparency to open-source software supply chains, can be used to bring transparency to open ML models, in terms of enabling model publishers to sign their models and prove properties about the datasets they use.

## 1. Introduction

In traditional software engineering, open-source code repositories hosted on platforms like GitHub are invaluable in enabling developers to build complex applications without having to write every component themselves. In machine learning, a similar paradigm is emerging: general-purpose (or *foundation*) models are trained and uploaded to model hubs such as Hugging Face[1] and Kaggle,[2] which users can then download and adapt for specific tasks. Crucially, this latter step is significantly cheaper than the former, and thus can be performed by entities who lack the resources needed to train a large foundation model.

While the models uploaded to these hubs are versioned and accompanied by *model cards* (Mitchell et al., 2019), this information must be taken at face value as there is nothing preventing the model publisher from lying about it. In other words, these claims are not *verifiable*; i.e., they are not accompanied by any kind of proof that they are true. Furthermore, while model cards typically do (or should) provide detailed information about aspects like intended usage, evaluation, and risks, they are often more vague about the training data, preprocessing, and training hardware.

Attacks in other ecosystems have motivated the creation and deployment of *transparency* solutions; e.g., the 2011 hack of the DigiNotar CA (Wolff, 2016) led to the creation of Certificate Transparency (CT).[3] CT was designed to protect users from misissued X.509 certificates by storing all issued certificates in globally accessible transparency logs; i.e., logs that are append-only and can be inspected by domain operators for potentially misissued certificates. Likewise, the 2020 software supply chain attack on SolarWinds (Newman, 2022) led to the creation of Sigstore (Newman et al., 2022),[4] a project designed to protect users from vulnerable or malicious software libraries using a similar approach to CT; i.e., having developers sign their code and store the signing metadata in a transparency log.

In the ML ecosystem, similar attacks are emerging: users of model hubs have been targeted with malware (Cohen, 2024; Wang et al., 2021), models trained to give misinformation on targeted prompts (Huynh & Hardouin, 2023; Bagdasaryan & Shmatikov, 2022), and models impersonating those from prominent organizations (Kiani, 2024). Going beyond models uploaded to hubs with malicious intent, there is also a clear need for model trainers to provide more transparency (Bommasani et al., 2024) into the models they create, as demonstrated by (proposed) regulation in both the US (David, 2024; Biden, 2023) and EU (2024); a growing number of lawsuits brought against model creators over their use of copyrighted work (Grynbaum & Mac, 2023; Vincent, 2023; Ho, 2024); and the demonstrated ability to *poison* training data in a way that biases the resulting model (Carlini et al., 2024). While some of this scrutiny

---

[1] Google [2] Google DeepMind. Correspondence to: Sarah Meiklejohn <s.meiklejohn@ucl.ac.uk>.

*Proceedings of the $42^{nd}$ International Conference on Machine Learning*, Vancouver, Canada. PMLR 267, 2025. Copyright 2025 by the author(s).

[1] https://huggingface.co/

[2] https://www.kaggle.com/models

---

[3] https://certificate.transparency.dev/
[4] https://www.sigstore.dev/

focuses on transparency around access to the models and what is and isn't synthetic data (i.e., data generated by a model), much of the focus is on the training data they use.

**Our contributions.** In this paper, we explore the topic of *model transparency* through the lens of supply chain security. We use this broad term to consider both concerns around training data *provenance*, which have been extensively considered (Longpre et al., 2024), and the less studied problems of ensuring model integrity and addressing other supply-chain risks.

Due to the urgency of deploying protections in this space, we propose solutions in this work that are designed to be performant and are built on simple and standardized cryptographic techniques. Concretely, we propose two distinct directions: first, in Section 5, we propose a simple but effective intervention, analogous to existing approaches like CT and Sigstore, in which publishers sign the models they upload to hubs. We have released this work as an open-source library and are working to integrate it into existing model hubs. Next, in Section 6 we propose applying existing *verifiable data structures* (e.g., Merkle trees) to the problem of model transparency. Specifically, we focus on how model trainers can commit to the data they use to train their models in a way that allows them to later prove the (non-)inclusion of specific data points in the set.

## 2. Alternative Views

Our view in this work is that the open ML model ecosystem has significant supply-chain risks, and that these risks are already being exploited; as such, we view this as a problem for which solutions urgently need to be designed and deployed. The most reasonable view in opposition to ours is that these are not the most urgent problems facing open ML models, and that furthermore (1) deploying solutions today would be premature given the relative instability of the tooling and the ecosystem as a whole and (2) any viable solutions are too costly to be considered practical. We hope that the model signing solution we put forward in this work helps to address this second concern, and argue that initial interventions can (and should) be adopted by model hubs rather than integrated into frameworks or other lower-level tooling, thus addressing the first concern as well.

## 3. Model Transparency

As mentioned already, we use the broad term *model transparency* when considering solutions addressing supply-chain risks for ML models. We consider both the need for interventions due to attacks that have already happened, as well as proposed or deployed interventions that may be effective in increasing transparency. We consider trans-

parency for supply chains themselves rather than for the outputs of deployed models (Narayanan & Kapoor, 2023).

### 3.1. Data provenance

Model cards (Mitchell et al., 2019) allow publishers to self-report information about their models, their intended uses, and how they were produced. The Foundational Model Transparency Index (Bommasani et al., 2024) provides a numerical score for foundation models based on how much data about the model and its components has been publicly disclosed. Both of these mechanisms rely on self-reported data, which cannot be assumed to be accurate for malicious publishers. Furthermore, even honest publishers may not want to reveal all information about all aspects of the model (e.g., the training data), so there is still value in making verifiable, privacy-preserving claims even with an accurate model card or other self-reported data. Likewise, researchers have proposed multiple standards for documenting and maintaining information about training datasets (Gebru et al., 2021; Luccioni et al., 2022; Pushkarna et al., 2022; Akhtar et al., 2024). Again, these works are complementary to our own in focusing on communication and documentation but not verifiability.

Longpre et al. (2024) provide an extensive exploration of the topic of data provenance, highlighting many problems that stem from poor due diligence and transparency around training data. Cen et al. (2023) argued that these harms are exacerbated by the current state of ML supply chains, in which the capabilities of models are not well specified and different components (e.g., models or datasets) interact with each other in unexpected and often unknown ways. Indeed, researchers have demonstrated the ease with which an attacker could poison training datasets (Carlini et al., 2024) and the attacks that can be carried out with poisoned training data (Gu et al., 2017) or pre-trained models (Gu et al., 2017; Kurita et al., 2020; Jiang et al., 2023). Longpre et al. also discuss many existing interventions for tracking data provenance, including both ones that offer some notion of verifiability and ones that do not.

In terms of verifiability, Choi, Shavit, and Duvenaud (2023) propose proofs of training data, in which a model trainer provides a training transcript to a verifier that is designed to convince them that a given model was obtained by running a specific training algorithm on a specific training dataset and hyperparameters. A similar goal was considered for stochastic gradient descent by Baluta et al. (2023), who defined forgery for intermediate model checkpoints and the conditions under which forgeries are (im)possible. Both of these approaches, however, require providing information such as the entire training dataset to the verifier. As such, they are not well suited to applications in which the model trainer would like to keep this information private.

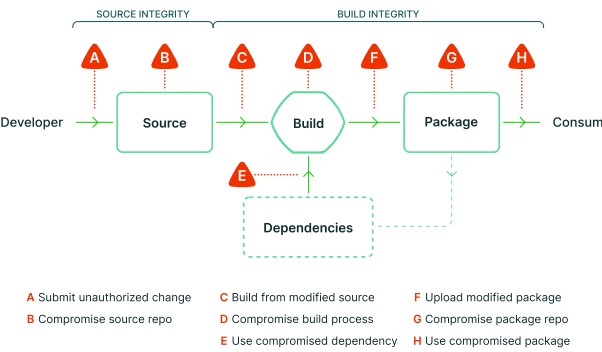

*Figure 1.* The different actors and phases in the software supply chain, along with the points at which malicious actions can be taken. This image was taken from https://slsa.dev/spec/v1.0/threats-overview.

A recent line of work has sought to overcome this limitation by using zero-knowledge proofs to prove the correctness of training without revealing the training data. Garg et al. (2023) and Tan et al. (2025) provide proofs for logistic regression, Eisenhofer et al. (2025) use SNARKs to provide proofs of training for both regression models and neural networks, and Abbaszadeh et al. (2024) provide a proof for deep neural networks. These protocols come with strong cryptographic guarantees but high overhead in terms of either large proof sizes or high prover runtime. There is thus currently no proof system that has demonstrated the ability to scale to large models while maintaining reasonable proof sizes and prover and verifier runtimes.

## 3.2. Model integrity

As we can see, significant concerns have already been raised by the community around issues of training data provenance and transparency. A more overlooked area, however, is *model integrity*, in terms of providing some assurance that the models published on model hubs are the ones that the model owner intended to publish, and in particular have not been tampered with by the model hub or some other malicious party. As we see below, the potential for such attacks in the current open model ecosystem is high (and in some cases attacks have happened already) due to the ease with which they can be carried out and the lack of deployed protections.

To understand these threats systematically, we can consider the software supply chain—as depicted in Figure 1—by way of analogy. We can think of the training data as corresponding roughly to a subset of the dependencies that go into producing a software artifact, the code for training as corresponding to the source code, the training itself as corresponding to the build process, and the final trained model as corresponding to a software artifact.

Several attacks of the types presented in Figure 1 have been demonstrated for the ML model supply chain in recent years. Researchers have observed the potential for attacks to be carried out at training time (Bagdasaryan & Shmatikov, 2021; Shumailov et al., 2021) (attack D), and there is at least one documented case of a real training process being compromised, with reporting in October 2024 indicating that ByteDance fired an intern for interfering with training (Belanger, 2024).

The concerns discussed around using poisoned, unintended, or otherwise harmful datasets and pre-trained models correspond to attack E (using a compromised dependency), but other attacks fall into this category as well. Several attacks have been demonstrated on PyTorch (The PyTorch Team, 2022; Young, 2022; Stawinski, 2024) and TensorFlow, two popular machine learning frameworks. Some of these attacks compromised the direct dependencies of the frameworks, while others exploited aspects of their continuous integration (CI) pipelines in ways that allowed for arbitrary code injection. The software supply chain of these frameworks was also investigated in depth by Tan et al. (2022), and Gao, Shumailov, and Fawaz (2025) demonstrated that these frameworks are vulnerable to new attacks due to their use in ML services, thus requiring an ML-specific threat model. Recent "tool poisoning" attacks on the Model Context Protocol (MCP) (Invariant Labs, 2025) demonstrated how unverified MCP tools can expose users to malicious packages, in addition to other risks. Finally, Langford et al. (2025) discussed how architectural backdoors—i.e., backdoors that rely solely on changing the model architecture—can be carried out in a variety of ways, ranging from poisoning training data or compromising the framework (attack E) to modifying the architecture definition file directly (attack F).

Uploading a new version of a model (attack F) or compromising the model repository (attack G) could easily be carried out by a malicious insider or model hub. In addition, because of the way models are deserialized, attacks are possible against models produced by honest publishers but that have been serialized using unsafe formats such as pickle (Milanov, 2024); moreover, attacks have targeted services designed to help users convert to safer formats (Hidden Layer, 2024). Researchers also discovered 1,681 leaked API tokens for Hugging Face (Lanyado, 2023), creating the potential for attackers to access the accounts of others and upload malicious models under their name. As observed by Jiang et al. (2023), the impact of this attack can vary significantly given that some maintainers have access to hundreds of different models.

Finding ways for a model consumer to use a compromised model (attack H) is even easier than compromising an existing model, as there are many ways for an attacker to con-

vince users to download their model. For example, Kiani reports namesquatting attacks on Hugging Face (2024), in which attackers try to make it appear as though their models are coming from a reputable organization, or are themselves well known models (e.g., Llama). Researchers have also uploaded models designed to provide misinformation on targeted topics (Huynh & Hardouin, 2023) and found over a hundred instances of malware on Hugging Face (Cohen, 2024), and Protect AI maintains a database with dozens of examples of unsafe or suspicious models.[5] While many articles and sources of documentation discuss the potential for attacks in an ecosystem in which users download and run arbitrary models (TensorFlow), there are few specific protections in place and traditional anti-virus solutions are often unable to catch these malicious models (McInerney, 2024).

## 4. Existing Solutions for (Model) Integrity

Preventing attacks on model integrity using technical solutions is difficult, if not impossible, given that many of the attacks are associated with longstanding problems that the security community has been unable to solve (e.g., stopping the spread of malware) or rely on social engineering rather than technical means (e.g., the use of namesquatting to convey trust, or finding leaked API tokens).

Instead, we can look to other domains for inspiration in terms of solutions that aim for the *detection* of misbehavior rather than its *prevention*. In particular, we consider Certificate Transparency (CT), a project that was created in response to the similar threat of certificate authorities (CAs) misissuing certificates for domains—either due to compromise or just poor due diligence—thus enabling a man-in-the-middle attack whereby an attacker could impersonate the operator of that domain. CT relies on two components: signatures created by CAs, which provide non-repudiable evidence that they did issue a given certificate, and transparency logs that contain all certificates that have been issued (by any CA). Crucially, CT does not *prevent* misissuance by promising that a given certificate is "good" if it appears in the log. Rather, it promises the ability for a domain operator to *detect* misissuance by inspecting the contents of the log, and thus hold accountable the CA who issued the certificate or otherwise act on this information. As we see in the next section, Sigstore (Newman et al., 2022) provides similar guarantees for open-source software, and can be easily adapted to work for open models as well.

Hugging Face, a popular model hub, has a deployed solution called *commit signing*,[6] wherein model publishers can sign their commits using GPG keys. As shown by Jiang et

al. (2023) and Schorlemmer et al. (2024), however, this approach has been adopted by fewer than 2% of the models on the hub. In software supply chains, PGP signatures are required for Java packages to be published in the Maven Central package registry. On the other hand, a similar lack of adoption as Hugging Face was observed when the Python Package Index (PyPI) offered the option to upload PGP signatures alongside artifacts, along with other issues such as a significant portion of signatures being unverifiable. These combined issues led PyPI maintainers to remove support for PGP signatures (Stufft, 2023).

Besides Sigstore, there are several other modern solutions for code signing and software supply chain transparency. In Sigsum,[7] users manage their own keys and record signatures to a key-usage transparency log. The Notary Project,[8] and more specifically the Notation CLI,[9] offer signing and verification of artifacts (primarily containers) in formats compliant with specifications set by the Open Container Initiative (OCI). Notary does not record signatures in a transparency log and requires key management. SCITT[10] is a framework for software supply chain security and transparency in which—like Sigstore—signatures are associated with identities and stored in a transparency log. Unlike Sigstore, however, SCITT mandates a specific format (CBOR Object Signing and Encryption, or COSE) for signatures; furthermore, their development is in an early stage and described as not suitable for production. Thus, as compared to these other solutions, we chose Sigstore due to its maturity, transparency, and the flexibility it offers in terms of formats and key management.

## 5. Integrity for ML Models

As a first step towards securing the open model ecosystem, we propose storing signing metadata in the Rekor transparency log; i.e., the log maintained by Sigstore. This protection is analogous to the one offered by Certificate Transparency and allows users to be sure that, when they download a given version of a model, they are seeing the canonical model at that version and not a targeted one; i.e., that they are seeing the same version as everyone else. If model publishers are also monitoring the contents of the log and making sure that all signing events for their models were intentional on their part, this protects users from models that have been tampered with or were otherwise maliciously crafted. This solution thus defends against attacks analogous to attacks F and G from Figure 1.

---

[5] https://protectai.com/insights/models

[6] https://huggingface.co/docs/hub/en/security-gpg

[7] https://www.sigsum.org/

[8] https://notaryproject.dev/

[9] https://github.com/notaryproject/notation

[10] https://datatracker.ietf.org/group/scitt/about/

## 5.1. Cryptographic notation

For a finite set $S$, $|S|$ denotes its size and $x \xleftarrow{\$} S$ denotes uniformly sampling a member from $S$ and assigning it to $x$. For an ordered list $L$, $L[i]$ denotes the $i$-th entry. $\lambda \in \mathbb{N}$ denotes the security parameter and $1^\lambda$ its unary representation. For $n \in \mathbb{N}$, $[n]$ denotes $\{1, \ldots, n\}$. By $y \leftarrow A(x_1, \ldots, x_n)$ we denote running algorithm $A$ on inputs $x_1, \ldots, x_n$ and assigning its output to $y$, and by $y \xleftarrow{\$} A(x_1, \ldots, x_n)$ we denote running $A(x_1, \ldots, x_n; R)$ for a uniformly random tape $R$. Our constructions in this and the next section rely on the discrete logarithm assumption, which says that it is hard to output $x$ given $g^x$ where $g$ is a generator of a group $G$ of prime order $p$ and $x \xleftarrow{\$} \mathbb{F}_p^*$, and the DDH (decisional Diffie-Hellman) assumption, which says that it is hard to distinguish between $(g, g^a, g^b, g^{ab})$ and $(g, g^a, g^b, g^c)$ for $a, b, c \xleftarrow{\$} \mathbb{F}_p^*$.

## 5.2. How Sigstore works

Our model signing solution builds on Sigstore (Newman et al., 2022), which was designed to provide transparency for open-source software supply chains. In addition, and in contrast to the commit signing solution from Hugging Face, Sigstore was designed to avoid key management on the part of developers.

Briefly, a typical usage of Sigstore is as follows. After creating a new software artifact, a developer can generate an ephemeral keypair for a digital signature scheme. They can then obtain a certificate from Sigstore's certificate authority, Fulcio, that binds the ephemeral public key to an OpenID Connect (OIDC) identity that the developer has demonstrated they control (typically by authenticating themselves to an approved identity provider, such as Microsoft or Google, and receiving an identity token in return that they can provide to Fulcio). To avoid the need for certificate revocation, certificates are typically short-lived; e.g., they are valid for only 10 minutes. After signing some metadata for the software (such as its hash) with the corresponding private key, the developer can then submit the metadata, certificate, and signature to Sigstore's transparency log, Rekor. Crucially, at this point the developer no longer needs to use the private signing key. Once accepted, Rekor returns to the developer a proof containing, among other objects, a Merkle inclusion proof of the entry; i.e., the sibling hashes along the path from the new entry to the root of the Merkle tree maintained in Rekor. The developer can then make a *bundle* of evidence available alongside their artifact so that users can verify both the signature and the fact that the relevant information was appropriately logged (and is thus available for inspection by external entities).

## 5.3. Model signing

We start by assuming that a model publisher has trained and saved a new version $M$ of a model mdl. This can be done using any number of frameworks; our solution is agnostic to the framework or the format of the final model.

We also assume that the publisher, with some identity id, is authorized to publish new versions of the model. To sign the model, the publisher follows the typical workflow for Sigstore (Newman et al., 2022) (Algorithm 1), which means: (1) obtaining an OIDC token tok; (2) forming an ephemeral keypair $(\mathsf{pk}, \mathsf{sk}) \xleftarrow{\$} \mathsf{KeyGen}(1^\lambda)$ and a proof of possession $\sigma_{\mathsf{pop}} \xleftarrow{\$} \mathsf{Sign}(\mathsf{sk}, \mathsf{id})$; (3) obtaining a certificate cert by submitting tok, pk, and $\sigma_{\mathsf{pop}}$ to Fulcio; and (4) running $\mathsf{Sign}(\mathsf{sk}, \mathsf{cert}, \mathsf{mdl}, M)$ to obtain a bundle. At this point the publisher can publish the bundle alongside $M$.

Upon encountering a model $M$ with an associated bundle, a model consumer can verify it using the regular Sigstore verification algorithm (Newman et al., 2022) (Algorithm 3).

In the open model ecosystem, publishers can upload bundles to model hubs alongside their models; indeed, this can be the default option when using the upload API for the hub. If the hub acts as an OIDC provider, then this can happen seamlessly as the model publisher needs to be authenticated with the hub anyway. These bundles can first be verified by the hub itself, who can optionally convey this "verified" status to users, before also being made available to users who may wish to perform verification themselves.

## 5.4. Implementation and evaluation

We implemented model signing in Python, on top of the `sigstore-python` library.[11] Our implementation is 4500 lines of code and available as an open-source library.[12]

Much of our code is devoted to serialization. In particular, both the model publisher and consumer(s) need to form a hash $h_{\mathsf{mdl}}$. If the model is a single file then this means hashing that file, and if the model is a directory then we hash its subdirectories and files (in alphabetical order, as determined by the full path of the object). For models that are saved as a single file, the signing metadata is stored in a file `<fname>.sig` (where 'fname' is the filename of the model). For models that are saved as a directory, it is stored in a file `model.sig` in the top-level directory. The library supports both SHA-256 and BLAKE2 as hash functions.

Moreover, the files for ML models can be large. Our code contains the option to hash these files naïvely, but also an optimized approach in which we break large files up into

---

[11] https://github.com/sigstore/sigstore-python
[12] https://github.com/sigstore/model-transparency/

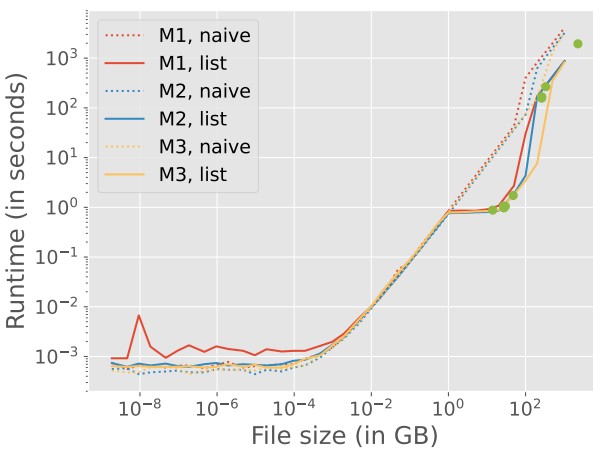

| Model | Size (in GB) | Time (in s) | |
|---|---|---|---|
| | | Hash | Total |
| gemma-2b | 14.03 | 0.88 | 1.36 |
| falcon-7b | 26.89 | 1.06 | 1.58 |
| Mistral-7B | 27.47 | 0.97 | 1.45 |
| Llama-3.1-8B | 29.93 | 1.05 | 1.48 |
| gemma-7b | 47.74 | 1.73 | 2.23 |
| Mixtral-8x22B | 261.93 | 157.26 | 157.97 |
| Llama-3.1-70B | 262.86 | 167.36 | 168.26 |
| falcon-180B | 334.39 | 266.03 | 266.81 |
| Llama-3.1-405B | 2275.95 | 1940.62 | 1941.23 |

*Table 1.* Averaged across five runs, the time required to hash and sign large open models using the list-based approach on M3.

*Figure 2.* Averaged over five runs and plotted on a log-log scale, the time, in seconds, required to hash a file of a size ranging from 1 B to 1 TB, on three different machines and using our two different approaches. The green dots represent the time to hash different large open models using the list-based approach on M3, as summarized in Table 1.

smaller chunks $C_1, \ldots, C_n$, and then hash these chunks in a list as $H(H(C_1)\| \ldots \|H(C_n))$. Our default chunk size is 1 GB, but our code contains the option to treat chunk size as a parameter.

We benchmarked our hashing code, using SHA-256 and both the naïve and list-based approaches, for file sizes ranging from 1 B to 1 TB, on three machines: (1) M1 with 24 vCPUs running on AMD EPYC 7B12 at 2.25 GHz and 96 GB of RAM; (2) M2 with 64 vCPUs running on AMD EPYC 7B13 at 2.45 GHz and with 120 GB of RAM; and (3) M3 with 128 vCPUs running on AMD EPYC 7B13 CPUs at 2.45 GHz and with 240 GB of RAM. We also benchmarked the full signing and verification workflows using ECDSA P256 (the default signature in Sigstore). The results are in Figure 2.

The naïve approach is initially faster, due to the orchestration overhead of the list-based approach, but gets significantly slower starting at 1 MB. This indicates the point at which disk performance becomes the main bottleneck. We can also see a "plateau" for the list-based approach after files become larger than 1 GB, which represents the point at which files start to get broken into smaller chunks. Finally, we see a sharp increase in the time required to hash files larger than the machine's available RAM. Prior to this threshold, Linux's disk cache allowed for portions of the file to be read from RAM instead of disk. Hashing the 1 TB file using the list-based approach takes 800 seconds (13.3 minutes) on the most powerful machine (M3). Compared to the cost of saving this file, which was 4,151 seconds (69 minutes), this represents a 19% overhead.

We also summarize in Table 1 the costs associated with hashing and signing a variety of large open models (as obtained from Hugging Face); we chose these ones as they are the largest open models available today. The reported model sizes include all files in the repository. The hashing costs are also depicted as green dots in Figure 2. As we can see, the costs of hashing these models—which are saved as directories—are entirely aligned with the costs of hashing a single large file. Furthermore, the additional costs of signing (which are dominated by the call to Rekor) range from 478 ms (35% overhead) to 602 ms (0.03% overhead), in line with the numbers reported for Sigstore (Newman et al., 2022). We did not include the time to verify in the table, but here the additional costs were consistently around 6 ms; these costs are much lower than for signing because verification happens entirely offline. We also measured the memory usage of signing each of these large models and found that peak memory usage ranged from 364.48 to 684.95 MB of RAM.

Overall, we can see that the costs of signing and verifying a model are dominated by the cost of hashing. Furthermore, the main bottleneck in hashing performance quickly becomes disk performance, suggesting that we would see similar performance using other hash functions.

## 6. Dataset Verifiability

Model signing is a useful first step in protecting users against malicious models, in terms of committing model publishers to their intent to publish a specific version of a model. On its own, however, it tells the user no information about what a model actually does or how it was produced; i.e., it does not address any concerns around data provenance (Longpre et al., 2024). We investigate this latter question in this section, in terms of addressing attack D from Figure 1.

In particular, we consider the problem of committing to the data that was used to train a model, in a way that allows a model trainer to later prove that a queried data point was or wasn't included in the training data; this could, for example, could be useful in allaying concerns about copyright infringement (David, 2024). Crucially, a model trainer must be able to prove these properties without revealing any other information about the training data, which is typically considered highly sensitive for models produced by large organizations. We leave open the potential for proving other interesting properties about the training data; e.g., that it does not counteract the guarantees of a differentially private training algorithm (Shamsabadi et al., 2024).

## 6.1. Cryptographic primitives

### 6.1.1. ZERO-KNOWLEDGE PROOFS

A zero-knowledge proof of knowledge for a relation $R$ consists of two algorithms (and an optional algorithm Setup to generate a *reference string* that is then input to the other algorithms): $\pi \xleftarrow{\$} \mathsf{Prove}(x, w)$ takes in an instance $x$ and a witness $w$ and outputs a proof $\pi$ that $(x, w) \in R$, and $0/1 \leftarrow \mathsf{Verify}(x, \pi)$ takes in the instance and the proof and outputs $1$ if the proof verifies and $0$ otherwise. The proof satisfies *zero knowledge* if a (PPT) simulator without knowledge of a witness can produce proofs that are indistinguishable from honest ones, and *knowledge soundness* if it is possible to extract (via a PPT extractor) a valid witness from any proof that verifies.

### 6.1.2. VERIFIABLE RANDOM FUNCTIONS (VRFS)

A verifiable random function (VRF) is a keyed pseudorandom function that enables efficiently verifiable proofs of correct evaluation. Formally, a VRF is comprised of four algorithms: (1) $\mathsf{pk}, \mathsf{sk} \xleftarrow{\$} \mathsf{KeyGen}(1^\lambda)$ outputs a public verification key and a secret evaluation key; (2) $y \xleftarrow{\$} \mathsf{Eval}(\mathsf{sk}, x)$ outputs the VRF evaluation of $x$ under the secret key $\mathsf{sk}$; (3) $\pi \xleftarrow{\$} \mathsf{Prove}(\mathsf{sk}, x, y)$ proves that $y$ is the correct evaluation of $x$ under $\mathsf{sk}$; and (4) $0/1 \leftarrow \mathsf{Verify}(\mathsf{pk}, x, y, \pi)$ checks the correctness of the evaluation $y$ of $x$ with respect to the public key $\mathsf{pk}$.

The security properties provided by a VRF are *pseudorandomness*, which says that even an adversary who picks the input $x$ should not be able to distinguish $\mathsf{Eval}(\mathsf{sk}, x)$ from random, and *unique provability*, which says that there should not be two different evaluations and proofs that verify for the same input $x$.

We use the VRF proposed by Melara et al. (2015), which is secure under the DDH assumption. This VRF operates in a group $G$ of prime order $q$, with generator $g$, and makes use of two associated hash functions: $H_1 : \{0, 1\}^* \to G$ and $H_2 : \{0, 1\}^* \to \mathbb{F}_q$. Within this context, the algorithms

are defined as follows: $\mathsf{KeyGen}(1^\lambda)$ picks $\mathsf{sk} \xleftarrow{\$} \mathbb{F}_q^*$, sets $\mathsf{pk} \leftarrow g^{\mathsf{sk}}$, and outputs $\mathsf{pk}, \mathsf{sk}$. Next, $\mathsf{Eval}(\mathsf{sk}, x)$ outputs $y \leftarrow H_1(x)^{\mathsf{sk}}$. $\mathsf{Prove}(\mathsf{sk}, x, y)$ picks $r \xleftarrow{\$} \mathbb{F}_q^*$, computes $s \leftarrow H_2(x, g^r, H_1(x)^r)$ and $t \leftarrow r - \mathsf{sk} \cdot s$, and outputs $\pi \leftarrow (s, t)$. Finally, $\mathsf{Verify}(\mathsf{pk}, x, y, \pi)$ outputs $1$ if $s = H_2(x, g^t \cdot \mathsf{pk}^s, H_1(x)^t \cdot y^s)$ and $0$ otherwise.

### 6.1.3. ACCUMULATORS

A cryptographic accumulator allows a (untrusted) prover to provide a succinct commitment to a set of data, in a way that allows it to later prove properties of the set against the commitment. In particular, we consider accumulators for which it is possible to prove that certain elements are or are not included in the set.

Formally, an accumulator $\mathsf{Acc}$ is comprised of five algorithms: (1) $\mathsf{state}, \mathsf{com} \xleftarrow{\$} \mathsf{Commit}((D_i)_i)$ outputs a commitment to an ordered list of elements $(D_i)_i$ and some internal state; (2) $\pi \xleftarrow{\$} \mathsf{ProveIncl}(\mathsf{state}, D)$ outputs a proof of inclusion of a given element; (3) $0/1 \leftarrow \mathsf{VerIncl}(\mathsf{com}, D, \pi)$ verifies the inclusion proof against the commitment com; (4) $\pi \xleftarrow{\$} \mathsf{ProveNonIncl}(\mathsf{state}, D)$ outputs a proof of non-inclusion of a given element; and (5) $0/1 \leftarrow \mathsf{VerNonIncl}(\mathsf{com}, D, \pi)$ verifies the non-inclusion proof against the commitment com. The main security property provided by an accumulator is *soundness*, which says that it should not be possible to provide a verifying inclusion proof for any elements not in the set, or a verifying non-inclusion proof for any elements in the set.

### 6.1.4. ZERO-KNOWLEDGE SETS

Accumulators provide the ability to efficiently prove (non-)inclusion against a commitment, but proofs may reveal information about the underlying set. A zero-knowledge set (ZKS) provides this additional privacy guarantee, in addition to the soundness provided by an accumulator.

Formally, a zero-knowledge set $\mathsf{ZKS} = (\mathsf{Commit}, \mathsf{Query}, \mathsf{Verify})$ is comprised of three algorithms: (1) $\mathsf{state}, \mathsf{com} \xleftarrow{\$} \mathsf{Commit}((D_i)_i)$ outputs a commitment to an ordered list of elements $(D_i)_i$ and some internal state; (2) $\mathsf{resp}, \pi \xleftarrow{\$} \mathsf{Query}(\mathsf{state}, D)$ outputs a response indicating whether or not $D$ is in the set and an associated proof of (non-)inclusion; (3) $0/1 \leftarrow \mathsf{Verify}(\mathsf{com}, D, \mathsf{resp}, \pi)$ verifies the proof against the commitment according to the response resp. In addition to soundness, the ZKS should provide $\mathcal{L}$-*privacy* (Chase et al., 2019), which says that proofs reveal no information beyond a defined leakage function $\mathcal{L}$.

## 6.2. Committing to training data

We represent training data as a series of elements $D_1, \ldots, D_n$. Each element can represent either a sin-

gle data point or a dataset (consisting of potentially many points).

Cryptographically, our goal is to (1) provide a compact commitment to these elements such that we can (2) efficiently prove the (non-)inclusion of specific elements in a way that (3) doesn't reveal anything about the (other) elements that were used. This third requirement means we need to go beyond the standard properties of cryptographic accumulators, which do not typically offer any hiding guarantees, and use a *zero-knowledge set* that supports non-membership proofs. Once a commitment is formed, we can imagine it being stored in an AIBOM (bill of materials) (Bennet et al., 2024) that is published alongside the model.

Our construction follows the approach established by verifiable data structures built for the purpose of key transparency, as is now deployed in WhatsApp (Lawlor & Lewi, 2023), and is particularly inspired by SEEMless (Chase et al., 2019). Because we consider only a static accumulator (i.e., the entire set to which we are committing is fixed in advance), we can simplify their construction while achieving the same soundness and privacy guarantees, as well as the same space efficiency.

This means we consider two underlying primitives: a verifiable random function VRF and an accumulator Acc. Our choice of VRF is presented in Section 6.1.2. Our accumulator is a Merkle tree, with entries ordered lexicographically to enable efficient proofs of non-inclusion. In more detail:

- Acc.Commit($(D_i)_{i=1}^n$) orders the list $(D_i)_i$ lexicographically. It then forms a Patricia Merkle tree over this list. Concretely, this means the $i$-th leaf hash is formed as $h_i \leftarrow H(i\|D_i)$, and the parent hash of two children $p.c_0$ and $p.c_1$, where $p$ denotes their common prefix, is computed as $H(p\|h_{p.c_0}\|h_{p.c_1}\|c_0\|c_1)$. It returns state containing the structure of the tree (the position and hashes for all nodes, and the underlying (ordered) list $(D_i)_i$) and com $\leftarrow h_{\text{root}}$.

- Acc.ProveIncl(state, $D$) provides the Merkle inclusion proof for $H(i\|D)$ (i.e., the data for siblings along the path from this leaf to the root of the tree, which includes their hash, prefix, and children suffixes), where $i$ is the index of $D$ in the list.

- Acc.VerIncl(com, $D$, $\pi$) verifies the Merkle inclusion proof; i.e., it recomputes the root hash $h_{\text{root}}$ given the sibling hashes, prefixes, and suffixes, and verifies that $h_{\text{root}} = $ com.

- Acc.ProveNonIncl(state, $D$) finds the node $z$ associated with the longest prefix $p$ of $D$, and its children $p.c_0$ and $p.c_1$. It outputs $z$, $h_z$, $p$, $c_0$, $c_1$, $h_{p.c_0}$, $h_{p.c_1}$, and the Merkle inclusion proof $\pi_z$ for $z$.

- Acc.VerNonIncl(com, $D$, $\pi$) verifies the inclusion of $z$ in the tree using $\pi_z$. It then checks that $h_z = H(p\|h_{p.c_0}\|h_{p.c_1}\|c_0\|c_1)$, that $p$ is a prefix of $D$, and that $p.c_0$ and $p.c_1$ are not prefixes of $D$. If all these checks pass then it outputs 1; otherwise, it outputs 0.

The soundness of this construction was proved by Chase et al. (Chase et al., 2019). With the underlying building blocks established, our zero-knowledge set construction achieves privacy by making the following modifications. First, entries in the accumulator are ordered lexicographically, which leaks information about what is or isn't in the set. Entries in the ZKS are instead ordered randomly, with the VRF used to compute the index at which each entry should be placed. Specifically, for entry $D$ we compute its new index $j$ as $H(\text{VRF.Eval}(\text{sk}, D_i))$. Second, proving (non-)inclusion in the accumulator can require giving out the data for neighboring leaf nodes, which leaks information. Instead of storing a plain hash at each leaf, we thus store a commitment to the entry and provide these commitments (which leak no information about the underlying entry) instead. Specifically, we store the value $H(D_i, r_i)$ for some random $r_i$. The algorithm for forming this type of commitment, ZKS.Commit, can be found in Figure 3.

### 6.3. Proving (non-)inclusion in training data

For a data element $D$, the model trainer can prove that $D$ was or was not part of the training data for their committed model using the underlying accumulator. In particular, they can identify the intended location $d$ of the data point in the data structure by computing the VRF, prove that this is the right location ($\pi_{\text{VRF}}$), and then prove the (non-)inclusion of the point at that location using the appropriate algorithms for the accumulator ($\pi_{\text{Acc}}$). This is summarized in Figure 3 as ZKS.Query. The verifier can then verify this proof, using ZKS.Verify (in Figure 3), based on the underlying algorithms for the VRF and the accumulator.

### 6.4. Ensuring the validity of a commitment

Thus far, we described how to prove the (non-)inclusion of specific data points against a commitment provided by a model trainer. This does not prove, however, that the commitment accurately represents the data that was actually used in training the model. Thus, the proofs provided against this commitment become meaningful only when we can be sure the commitment was computed correctly.

As discussed in Section 3, a recent line of work has focused on providing proofs of training data (Choi et al., 2023; Baluta et al., 2023; Garg et al., 2023; Eisenhofer et al., 2025; Abbaszadeh et al., 2024; Tan et al., 2025). This work could be leveraged here by providing either a non-private proof of training data to an entity who is trusted to see the data

---

$\underline{\mathsf{ZKS.Commit}(D_1, \ldots, D_n)}$
$\mathsf{pk}, \mathsf{sk} \overset{\$}{\leftarrow} \mathsf{VRF.KeyGen}(1^\lambda)$
$L \leftarrow \vec{\varepsilon}$
for all $i \in [n]$ :
$\quad d_i \leftarrow H(\mathsf{VRF.Eval}(\mathsf{sk}, D_i))$
$\quad r_i \overset{\$}{\leftarrow} \{0, 1\}^\lambda$
$\quad h_i \leftarrow H(D_i, r_i)$
$\quad L[d_i] \leftarrow h_i$
$\mathsf{state_{Acc}}, \mathsf{com} \overset{\$}{\leftarrow} \mathsf{Acc.Commit}(L)$
$\mathsf{state} \leftarrow (\mathsf{state_{Acc}}, \mathsf{sk}, (D_i, r_i)_i)$
return $\mathsf{state}, (\mathsf{pk}, \mathsf{com})$

$\underline{\mathsf{ZKS.Query}(\mathsf{state}, D)}$
$d \leftarrow \mathsf{VRF.Eval}(\mathsf{sk}, D)$
$\pi_{\mathsf{VRF}} \overset{\$}{\leftarrow} \mathsf{VRF.Prove}(\mathsf{sk}, D, d)$
if $(D \in (D_i)_i)$
$\quad \pi_{\mathsf{Acc}} \leftarrow \mathsf{Acc.ProveIncl}(d)$
$\quad$ return $1, (d, \pi_{\mathsf{VRF}}, \pi_{\mathsf{Acc}})$
else
$\quad \pi_{\mathsf{Acc}} \leftarrow \mathsf{Acc.ProveNonIncl}(d)$
$\quad$ return $0, (d, \pi_{\mathsf{VRF}}, \pi_{\mathsf{Acc}})$

$\underline{\mathsf{ZKS.Verify}((\mathsf{pk}, \mathsf{com}), D, \mathsf{resp}, (d, \pi_{\mathsf{VRF}}, \pi_{\mathsf{Acc}})}$
$b_{\mathsf{VRF}} \leftarrow (\mathsf{VRF.Verify}(\mathsf{pk}, D, d, \pi_{\mathsf{VRF}}) = 1)$
if $(\mathsf{resp} = 0)$
$\quad b_{\mathsf{Acc}} \leftarrow (\mathsf{Acc.VerIncl}(\mathsf{com}, d, \pi_{\mathsf{Acc}}) = 1)$
else
$\quad b_{\mathsf{Acc}} \leftarrow (\mathsf{Acc.VerNonIncl}(\mathsf{com}, d, \pi_{\mathsf{Acc}}) = 1)$
return $b_{\mathsf{VRF}} \wedge b_{\mathsf{Acc}}$

---

*Figure 3.* Algorithms for our zero-knowledge set, assuming an underlying accumulator Acc and VRF VRF.

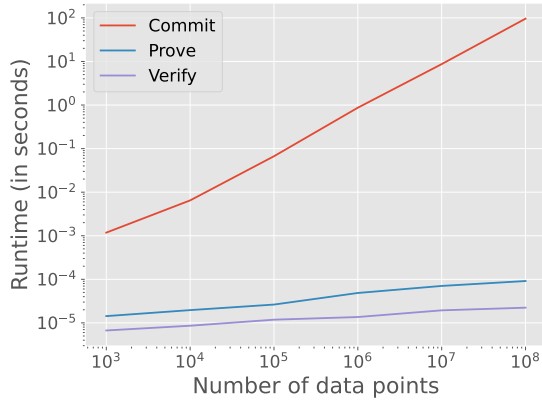

*Figure 4.* Averaged over ten runs and plotted on a log-log scale, the time, in seconds, to commit to and prove and verify inclusion in a data registry of a given size, ranging from 1000 to 100 million entries.

but not collude with the model trainer (e.g., a regulator), or publishing a zero-knowledge proof that anyone could verify. Another approach would be to train a model in an environment equipped with trusted execution environments (TEEs), which could be used to form the commitment and attest to its correctness. We leave the development of these and other potential solutions as interesting future work.

### 6.5. Evaluation

To benchmark the costs associated with this type of training data commitment, we use the available Rust code for the Parakeet verifiable registry (Malvai et al., 2023). We measured the costs of computing a commitment and proving and verifying against it for datasets ranging from 1000 to 10 billion data points. The results are in Figure 4.

As expected, Commit is orders of magnitude more expensive than proving or verifying (non-)inclusion: the time required for these operations is still on the order of tens of microseconds even for data structures with 100 million entries, whereas Commit requires 96 s to form a data structure of this size. We could not scale further using an in-memory data structure, but if we switch to using a persistent storage layer it seems likely we could scale to billions or tens of billions of entries (indeed, Parakeet was developed for use within WhatsApp, which has two billion users).

We chose such a wide range because we believe it accurately reflects the many potential use cases of this approach: fine-tuning a foundation model to output images in a specific visual style, for example, might use only hundreds of additional images. In these cases the party performing the fine-tuning could prove that they hadn't introduced any new copyrighted images and leave the scrutiny of the (much larger) foundation model to other parties or processes. On the other hand, smaller image models might be trained on well known datasets like CIFAR-10 (which has 50K rows)[13] or MNIST (60K rows),[14] while larger datasets like YouTube-Commons (400K rows)[15] are used for fine-tuning language models for Q&A tasks.

## 7. Conclusions and Future Work

In this work, we identify the risks posed by ML model supply chains today and demonstrate two first steps in securing these supply chains, in terms of signing models before they are published on hubs and committing to training data in such a way that the trainer can later prove the (non-)inclusion of specific data points. Given the preliminary nature of our work, there is a wide variety of future work, including developing new approaches that ensure the model trainer forms an accurate commitment to the training data and mapping out the supply-chain risks associated with more complex agentic AI systems.

---

[13] https://huggingface.co/datasets/cifar10
[14] https://huggingface.co/datasets/mnist
[15] https://huggingface.co/datasets/PleIAs/YouTube-Commons

## Acknowledgements

We are grateful to our anonymous reviewers for their helpful feedback, and to Vincent Roseberry, Meg Risdal, Nesh Devanathan, and others at Kaggle for working with us to integrate model signing into their hub.

## Impact Statement

As outlined throughout, verifiable solutions for model integrity and data provenance have the potential to positively impact the security and transparency of the open model ecosystem. In particular, they have the potential to protect users who download and run models from hubs and offer reassurance to interested parties (regulators, content creators, etc.) about how their data is—or is not—being used. We stress that the solutions offered in this paper do not yet achieve this potential, and that achieving the full potential will require careful consideration to ensure that further solutions provide meaningful information rather than becoming onerous or a check-box exercise that provides a false sense of transparency.

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
