# OpenReview forum: "Position: Machine Learning Models Have a Supply Chain Problem"
_ICML.cc/2025/Position_Paper_Track — ICML 2025 Position Paper Track poster_

### Official Review · Reviewer_qs4C · 2025-03-10

**Significance:** 2
**Argument Clarity:** 2
**Rating:** 3
**Confidence:** 3

**Questions:**

1) What are the risks that Sigstore can completely or partially mitigate? What are its limitations?

2) What are the requirements to apply the training dataset verifiability method proposed in Section 5 and in which practical contexts can it be applied?

3) Following the previous question, it would be good to know more about the robustness of the method against minor manipulations on the training data points to evade detection.


***************** POST-REBUTTAL COMMENTS **********************
Thank you very much for your responses and clarifications. They've been very helpful and clarified some of my concerns. Thus, I'm increasing my score.

**Discussion Potential:**

3

**Paper Summary:**

The paper discusses the security risks of machine learning models throughout the supply chain. The authors advocate for the analysis of model transparency, considering two main problems: model integrity and data provenance. In the first case, the authors propose Sigstore, a solution designed to bring transparency for open-source software, as a solution to check open ML models integrity, bringing more transparency. Secondly, the authors propose a solution based on verifiable data structures for proving the non-inclusion of specific data points in the training dataset of ML models, helping, for example, to detect copyright infringements.

**Position:**

Yes

**Position In Title:**

Yes

**Related Work:**

3

**Strengths And Weaknesses:**

Strengths:
+ The paper does a very nice analysis of the risks and vulnerabilities present in the supply chain for ML models, which can lead to different types of attacks. This is a very relevant problem where many challenges remain unsolved. The position of the authors advocating for model transparency can be interesting in some practical scenarios, especially for open-source ML models.

+ Related to the previous point, Section 2 is very nicely written and provides a good understanding of the risk present in the supply chain and a nice review of the research literature.

+ Sigstore appears a reasonable choice to check the integrity of ML models, as supported by the experiment described in Figure 2, showing the scalability of the approach considering the size of current ML models.


Weaknesses:
+ It would be good to understand why Sigstore is a more suitable approach for checking model integrity compared to other available alternatives.
+ Related to the previous point, Section 2 discusses the risks very nicely, however, a more thorough discussion of the risks that the application of Sigstore can mitigate, its coverage and limitations is missing.
+ In Section 5, it is unclear to me in which practical scenarios the training dataset verifiability method proposed can be applied and what are the requirements to make this a successful approach. On the other hand, I wonder up to what extend this solution is robust to, for example, cases where the entity collecting the datasets slightly manipulates the data to avoid being detected and accused of copyright infringement. It seems to me that the method can be very brittle in the presence of an adversary.

**Support:**

2

---

> ### Author Rebuttal · Authors · 2025-03-28
>
> Thanks for your review! We'll answer your questions in turn.
>
> > 1. What are the risks that Sigstore can completely or partially mitigate? What are its limitations?
>
> Please see our response to Reviewer w1v2’s second question for the first part of this question. For the second part, model signing is really a first step in that it provides a basic promise about a model: that these weights are the ones the model publisher intended. Following ideas that are already being used in the software supply chain community to provide software bills of materials (SBOMs), we are also exploring extending our model signing solution to signing AIBOMs that contain information like the frameworks, pre-trained models, or data that went into training; this extended solution would further address risks around compromised dependencies. We are happy to add a discussion of this to Section 5 or 6 of the paper.
>
> > 2. What are the requirements to apply the training dataset verifiability method proposed in Section 5 and in which practical contexts can it be applied?
>
> Unlike our model signing solution, the approach we propose for dataset verification is relatively preliminary; as such, we acknowledge that there are many practical considerations that would need to be addressed before deploying this or any other solution. As one example, we consider scalability. Data structures like the one we use have shown the ability to scale to tens of billions of entries (the current data structure used in WhatsApp key transparency contains ~90 billion entries), but large foundation models are typically trained on orders of magnitude more data than this, although some image models may use training data with a more comparable number of points. As such, the use case we highlight in Section 5.4 of proving properties of the data used in fine-tuning might be a more reasonable context to target rather than full model training.
>
> > 3. Following the previous question, it would be good to know more about the robustness of the method against minor manipulations on the training data points to evade detection.
>
> One option to improve robustness is to use a perceptual hash rather than a cryptographic hash at the leaves of the verifiable data structure. This would make the protocol more robust to these types of manipulations, because the hashes would match even if the image used in training had undergone minor manipulations as compared to the queried one, at the cost of incurring false positives in the form of users being told their data was used in training when it actually wasn’t (because perceptual hashes are more prone to collisions). Again, this demonstrates the many practical considerations and trade-offs that would need to be considered before deploying any solution for dataset transparency.
>
> > Weaknesses: It would be good to understand why Sigstore is a more suitable approach for checking model integrity compared to other available alternatives.
>
> Please see our response to Reviewer w1v2’s first question for our proposed way to expand our discussion of alternative approaches.

---

### Official Review · Reviewer_w1v2 · 2025-03-12

**Significance:** 3
**Argument Clarity:** 3
**Rating:** 3
**Confidence:** 4

**Questions:**

1. How does Sigstore compare to other potential solutions for ML supply chain security in terms of robustness, scalability, and ease of adoption?

2. Can the authors provide a more detailed analysis of how Sigstore mitigates each of the attacks outlined in Figure 1 and Section 2?

**Discussion Potential:**

2

**Paper Summary:**

This paper argues that the current open ecosystem for machine learning (ML) models contains significant supply chain risks, including model replacement attacks, training on restricted or poisoned data, and vulnerabilities in underlying frameworks. The authors explore how Sigstore, a transparency solution designed for software supply chains, can be applied to the ML model supply chain to enhance security. They propose using Sigstore for signing models at different development stages and proving dataset properties, providing a detailed evaluation of its applicability.

## Update after rebuttal

While the paper makes a timely and valuable contribution by addressing the emerging risks in ML model supply chains and proposing the use of Sigstore as a mitigation strategy, I remain moderately positive but not strongly convinced. The authors have responded constructively to concerns around comparative analysis and robustness, and they plan to expand their discussion of alternative solutions like commit signing, Notary, and SigSum, which helps address some of the paper’s gaps. Their explanation of how Sigstore defends against specific supply chain attacks (F and G) adds helpful clarity, although the coverage remains limited in scope. The planned presentation improvements, such as visualizing Sigstore’s integration into the ML pipeline, will also improve accessibility. However, many of these enhancements were not present in the original submission, and the core arguments could benefit from deeper technical rigor and broader evaluation.

**Position:**

Yes

**Position In Title:**

Yes

**Related Work:**

2

**Strengths And Weaknesses:**

Strengths:

* The paper addresses a timely and important issue—the security risks in ML model supply chains—which is highly relevant to the broader ML and cybersecurity communities.

* It presents a discussion of different supply chain threats specific to ML models, grounding its claims in real-world risks and attacks.

* The proposed use of Sigstore is motivated, and the evaluation provides useful insights into its feasibility for ML models.

Weaknesses:

* Insufficient comparison with existing solutions:

The discussion of alternative solutions (Section 3) is somewhat limited. A deeper exploration of existing approaches in software supply chain security and their applicability (or lack thereof) to ML models would strengthen the argument for Sigstore.

The paper would benefit from a more detailed analysis of why existing software supply chain methods cannot be easily adapted to ML supply chains.

* Limited evaluation of Sigstore’s robustness against supply chain attacks:

While the paper provides a strong case for Sigstore’s applicability, it lacks an in-depth discussion of how well it defends against the specific attacks outlined in Figure 1 and Section 2.

A qualitative or quantitative comparison with alternative solutions could provide better context on Sigstore’s relative strengths and weaknesses.

* Presentation improvements:

The paper could be made more accessible by incorporating more figures or diagrams illustrating existing solutions and their limitations.

Providing a visual breakdown of how Sigstore integrates with the ML model development and deployment pipeline would enhance clarity for readers unfamiliar with supply chain security.

**Support:**

3

---

> ### Author Rebuttal · Authors · 2025-03-28
>
> Thanks for your review! We’ll answer your questions in turn.
>
> > 1. How does Sigstore compare to other potential solutions for ML supply chain security in terms of robustness, scalability, and ease of adoption?
>
> We will definitely add an expanded discussion of alternative solutions to Section 3. For ML security specifically, the only other solution we are aware of is commit signing. To elaborate on our comparison in the paper, commit signing is more scalable (as we describe in our response to Reviewer 46L3’s first question) but arguably much harder to adopt due to the need for developers to manage GPG keys. Furthermore, it is less robust due to the fact that it provides integrity (albeit for commits, not for models) but not transparency. We elaborate on why this is a meaningful distinction in our response to your next question.
>
> In considering other solutions for the software supply chain that could be similarly adapted to the ML setting, there are several relevant projects (e.g., SigSum, which provides transparency for key usage, and Notary / Notation, which provides a way to sign software artifacts such as containers) and academic publications. We are happy to add an in-depth comparison with these in the paper; briefly, we chose Sigstore due to the advantages it offers in terms of (a lack of) key management, transparency, and a distributed root of trust. As a more practical consideration, Sigstore is also the most mature project; e.g., it is in general availability and offers an SLO of 99.5% availability. These considerations have been critical in our work to integrate model signing into a real model hub.
>
> > 2. Can the authors provide a more detailed analysis of how Sigstore mitigates each of the attacks outlined in Figure 1 and Section 2?
>
> We will add text to the start of Section 4 to make this clear. The attacks mitigated by our model signing solution are attacks F and G, in which an attacker uploads a tampered version of the model to a hub. This could happen either due to the attacker’s ability to tamper with the model in transit (which would not be detected if the model is not signed) or due to the attacker compromising someone else’s account and uploading a signed model without their knowledge. This latter attack demonstrates why transparency is crucial: the signature produced by the attacker is valid (meaning any integrity checks would pass), but a model publisher monitoring the transparency log would notice a signing event that they did not intend and realize that their credentials must have been compromised.
>
> > Weaknesses: Presentation improvements
>
> Thanks for the suggestions! We agree that these more visual elements would be helpful, and will incorporate them into the paper to the extent that space allows.

---

### Official Review · Reviewer_46L3 · 2025-03-14

**Significance:** 4
**Argument Clarity:** 4
**Rating:** 5
**Confidence:** 3

**Questions:**

1. Presumably, hugging face implemented commit signing instead of model signing because the latter was more complex to implement. What exactly were the main hurdles with traditional ways of implementing model signing in a pre-Sigstore model?
2. What scaling hurdles would Sigstore signing primarily face when dealing with models even larger than Llama-3.1-405B?

**Discussion Potential:**

4

**Paper Summary:**

Machine learning has a supply chain problem. We can use approaches from traditional software supply chain security to secure ML supply chains. For example, we can use Sigstore to sign models and use novel methods to prove properties about datasets used by models as well.

**Position:**

Yes

**Position In Title:**

Yes

**Related Work:**

4

**Strengths And Weaknesses:**

Strengths:
- Very clear position and innovative solution
- I was impressed by the level of analysis around performance -- such as checking times to perform the algorithms and plotting them

Weaknesses:
- Cryptographical notation is quite dense for those who are not familiar with it. Even though you couldn't include the GitHub repo link, it would have been good to include some real code in the appendix for some of the algorithms.

**Support:**

4

---

> ### Author Rebuttal · Authors · 2025-03-28
>
> Thanks for your review and your support! We'll answer your questions in turn.
>
> > 1. Presumably, hugging face implemented commit signing instead of model signing because the latter was more complex to implement. What exactly were the main hurdles with traditional ways of implementing model signing in a pre-Sigstore model?
>
> If we ignore the benefits of using Sigstore in terms of transparency and (lack of) key management and focus just on signing, signing a git commit is more efficient than signing the model due to the significant overhead of hashing the model. Commit signing avoids this overhead because, unlike a hash of the model, the commit has already been computed and is readily available. On the other hand, commit signing has to be done every time there is any change to the repository (even if it’s just to the documentation), whereas model signing can happen only when the model itself changes.
>
> > 2. What scaling hurdles would Sigstore signing primarily face when dealing with models even larger than Llama-3.1-405B?
>
> Section 4.4 demonstrates the extent to which hashing the model is (by far) the most expensive operation involved in our solution. This cost will only grow with larger models and we expect it to remain the largest scaling hurdle. All users engaging in these operations, however, are also downloading or saving and uploading the model, and we expect these other costs to remain dominant as they also grow with the size of the model (e.g., we estimate the cost of hashing the model to be ~5% of the cost of downloading it). We also plan to support a streaming version of model hashing, in which the model is hashed as it is downloaded, in a future release of the model signing library.
>
> > Cryptographical notation is quite dense for those who are not familiar with it. Even though you couldn't include the GitHub repo link, it would have been good to include some real code in the appendix for some of the algorithms.
>
> Yes, we agree :). We will link to the GitHub repository in the final version of the paper and otherwise do what we can to ease the notational burden.

---

### Official Review · Reviewer_spoK · 2025-03-14

**Significance:** 2
**Argument Clarity:** 3
**Rating:** 3
**Confidence:** 3

**Questions:**

- While your approach allows for dataset verification, how can we enforce data transparency without violating privacy laws?
- Could an attacker circumvent the signing process by creating convincing fake models with seemingly valid signatures?

**Discussion Potential:**

3

**Paper Summary:**

This paper discusses the supply chain vulnerabilities in the machine learning (ML) ecosystem. First, the authors introduce how publicly available ML models, particularly those hosted on platforms like Hugging Face and Kaggle, can be compromised in various ways. Second, the authors introduce the primary risks include: Model Attack, Data Poison, Misinformation and Dependency Attacks. Third, the author prepose cryptographic signing mechanisms for ML models to ensure Integrity, Provenance and Transparency and evaluate the method.
## update after rebuttal
remain the score, since the author didn't fully address the comments.

**Position:**

Yes

**Position In Title:**

Yes

**Related Work:**

2

**Strengths And Weaknesses:**

Strengths:
- The  security of ML supply chains is a critical and important issue in the ML real world application.
- The paper thoroughly explains different attack vectors in ML pipelines and provides real-world examples of past security breaches.
- The authors evaluate their approach and benchmark its performance, demonstrating its feasibility.

Weakness:
-  Deployment: The paper does not deeply explore how adoption barriers (e.g., industry standards, incentives, regulatory challenges) could affect real-world deployment.
- Data Integrity: The proposed solution does not fully prevent training data poisoning; it only allows post-hoc verification. More discussion on proactive prevention methods would strengthen the argument.
- While the paper discusses model impersonation, it does not fully address identity verification challenges.

**Support:**

3

---

> ### Author Rebuttal · Authors · 2025-03-28
>
> Thanks for your review! We'll answer your questions in turn.
>
> > While your approach allows for dataset verification, how can we enforce data transparency without violating privacy laws?
>
> We are not experts in privacy law, but our approach is designed to avoid revealing any information about the dataset beyond the point being queried (which, combined with rate limiting, makes it difficult for any one party to learn too much about the dataset). Furthermore, data transparency actually supports privacy laws like the GDPR in allowing data subjects to be aware of which third parties are using their data.
>
> > Could an attacker circumvent the signing process by creating convincing fake models with seemingly valid signatures?
>
> An attacker cannot create valid signatures for a model they don’t own, because by signature unforgeability they won’t be able to obtain a valid OIDC token (unless they are colluding with the token provider, which is outside the scope of our threat model; we will state this explicitly in the paper). They can easily create valid signatures for a model they do own, and these models can impersonate reputable models using techniques like typosquatting (if this is what you mean by models being “convincing fakes”). In other words, model signing provides an important base layer of defense but does not defend against many of the attacks we discuss in Section 2.2, as we discuss in our response to Reviewer w1v2’s second question. We hope this answers your question and will clarify the scope of the model signing solution in the paper.
>
> > Weaknesses
>
> We agree that preventative measures would be even better and that preventing model impersonation is an important problem. For now, we are matching the state of the art in terms of what other fields have been able to achieve; i.e., CT is also a post-hoc detection mechanism, and typosquatting is an urgent but largely unsolved problem in software package registries as well. We would be happy to add more details about our integration into a real model hub into the paper to give some insight into the adoption barriers we have already encountered.

---

> > ### Comment · Reviewer_spoK · 2025-04-05
> >
> > I have read the author's response and I would keep my score.

---

### Decision · Program_Chairs · 2025-04-30

**Decision:**

Accept (poster)

**Comment:**

This paper focuses on the ML model supply chain, and how hosted models can have various risks. These risks include attacks like data poisoning. The authors propose cryptographic solutions to alleviate these problems, including Sigstore.

This is an important and interesting problem, which the authors explore quite well. They cover a number of different attack vectors combined with real-world examples. Some weaknesses include that the proposal could have been compared more with alternatives, which the authors said they would elaborate on with an in-depth comparison in the final paper. Additionally, it is unclear whether it is that controversial a position to say that the ML supply chain is vulnerable -- this is pretty well known and accepted by the community.